# HIV-1/HBV Coinfection Accurate Multitarget Prediction Using a Graph Neural Network-Based Ensemble Predicting Model

**DOI:** 10.3390/ijms24087139

**Published:** 2023-04-12

**Authors:** Yishu Wang, Yue Li, Xiaomin Chen, Lutao Zhao

**Affiliations:** 1School of Mathematics and Statistics, University of Science and Technology Beijing, Beijing 100083, China; 2Center for Energy and Environmental Policy Research, Beijing Institute of Technology, Beijing 100081, China

**Keywords:** HIV-1/HBV coinfection, graph neural network, gradient boosting decision tree, multitarget prediction, drug discovery

## Abstract

HIV and HBV infection are both serious public health challenges. There are more than approximately 4 million patients coinfected with HIV and HBV worldwide, and approximately 5% to 15% of those infected with HIV are coinfected with HBV. Disease progression is more rapid in patients with coinfection, which significantly increases the likelihood of patients progressing from chronic hepatitis to cirrhosis, end-stage liver disease, and hepatocellular carcinoma. HIV treatment is complicated by drug interactions, antiretroviral (ARV) hepatotoxicity, and HBV-related immune reconditioning and inflammatory syndromes. Drug development is a highly costly and time-consuming procedure with traditional experimental methods. With the development of computer-aided drug design techniques, both machine learning and deep learning have been successfully used to facilitate rapid innovations in the virtual screening of candidate drugs. In this study, we proposed a graph neural network-based molecular feature extraction model by integrating one optimal supervised learner to replace the output layer of the GNN to accurately predict the potential multitargets of HIV-1/HBV coinfections. The experimental results strongly suggested that DMPNN + GBDT may greatly improve the accuracy of binary-target predictions and efficiently identify the potential multiple targets of HIV-1 and HBV simultaneously.

## 1. Introduction

HIV-1 is the human immunodeficiency disease, or AIDS virus type 1, which is currently the dominant strain in the global epidemic. HIV remains a major global public health problem, claiming approximately 40.1 million lives to date [1,2,3,4,5,6]. Hepatitis B virus, or HBV, is one of the smallest DNA viruses known to infect humans but is also one of the most difficult-to-cure viruses. It is well known that HIV and HBV share similar transmission routes, including sexual transmission, contaminated needles, transmission from mother to child, and the therapeutic use of blood [7]. Disease progression is more rapid in patients with coinfection, which significantly increases the likelihood of patients progressing from chronic hepatitis to cirrhosis, end-stage liver disease, and hepatocellular carcinoma. Moreover, HIV treatment is complicated by drug interactions, antiretroviral (ARV) hepatotoxicity, and HBV-related immune reconditioning and inflammatory syndromes [8,9]. Some related studies have demonstrated that liver disease caused by coinfection of HIV and HBV has become the second leading cause of death from HIV [10,11]. Moreover, HIV requires lifelong medication; meanwhile, patients with HIV infection have low immunity, while HBV infection can lead to a corresponding immune response. Therefore, for coinfection patients, the simultaneous treatment of both HIV and HBV is very complicated. Many researchers have noted that the most effective therapy is two of the three drugs used in highly effective combined antiretroviral therapy (HAART) for HIV that act against HBV [12,13,14]. However, drug toxicity, liver and kidney damage, osteoporosis, and other side effects should be considered in drug combinations. For example, the presence of HBV resistance mutations upon initiation or during antiretroviral therapy (ART) in HIV-coinfected patients is a common and inevitable treatment response [15]. Therefore, in view of the above factors, the discovery of inhibitors with dual antiviral effects and multiple targets is important and a unique strategy for the treatment of HIV/HBV coinfection.

The conventional strategies to obtain the properties of a given molecule usually require a series of complicated biochemical reactions, and with accumulated chemical molecules and rapidly emerging novel molecules, experimental methods become an impossible mission to determine a specific property of all molecules [16,17]. Machine learning methods have proven to be useful in multiple areas of drug discovery by calculating the quantitative structure–activity relationship (QSAR) models based on the molecules’ three-dimensional structures [18,19,20,21], including support vector machine (SVM) [22], random forest (RF) [23], naive Bayes (NB) [24], etc. In recent years, deep learning algorithms have rapidly developed and become very successful in various biochemical prediction areas. Deep learning with the use of deep neural networks may replace conventional machine learning algorithms and achieve performance improvements. However, standard CNN [25] and RNN [26] networks cannot handle feature representations such as nonsequentially ordered graph inputs. In view of the three-dimensional molecular properties, graph neural networks (GNNs) have achieved one of the major breakthroughs in detecting interatom connections. Various GNN subtypes have achieved efficiency in capturing internode relationships, such as graph convolutional network (GCN) [27,28], gated graph neural network (GGNN) [29], and direct message passing neural network (DMPNN) [30].

Specially, MPNNs [30] are a framework for learning local and global features from irregular forms of data (especially in molecular predictions), which are invariant to permutations. This network performs iterative massaging operations on each object and its neighbors, then obtains the final output from all messages regardless of their orders. In chemical prediction tasks, molecular characteristics can be learned directly from molecular graphs without being affected by isomorphism of graphs. Furthermore, in view of the outstanding performance of machine learning algorithms in target prediction of small molecules, we attempted to replace the output layer by the supervised learning method to improve the prediction performance, which has been effective in the following results. Particularly, one well-established machine learning algorithm, gradient boosting decision tree (GBDT) [31], is an efficient iterative decision tree algorithm, which constructs a set of weak learning machines (trees) and sums up results of multiple decision trees as the final predictive output. This algorithm combines the decision tree and integration idea effectively.

In this study, we proposed a graph neural network-based molecular feature extraction model by integrating one optimal machine learning classifier (by comparing the supervised learning ability with five-fold cross-validations), GBDT, to fish multitarget anti-HIV-1 and anti-HBV therapy. By comparing three different graph neural networks, GCN, GGNN, and DMPNN, in binary-target classification tasks with five-fold cross-validation, the DMPNN + GBDT ensemble model was adopted as the multitarget prediction model. To verify the application of this model, DMPNN + GBDT was employed to predict the potential multitargets of 22 approved HIV-1 drugs and 8 approved HBV drugs, as well as 10 compounds known to be active against at least 1 of the HIV-1 (/HBV) targets but not active against any of the HBV (/HIV-1) targets. The predicted results demonstrated eight approved drugs, which have been verified by the previous references, and six new compounds to be potential HIV-1/HBV coinfection multitarget inhibitors, which have been further confirmed by molecular docking simulations. Therefore, our study indicated that graph neural network-based multitarget prediction from molecular structures could potentially be applied to the discovery of HIV-1/HBV co-inhibitors.

## 2. Results

The workflow is shown in Figure 1. In this study, we first evaluated the performance of conventional machine learning methods including support vector machine (SVM) [22], random forest (RF) [23], naive Bayes (NB) [24], extreme gradient boosting algorithm (XGBoost) [32,33], and gradient boosting decision tree (GBDT) [31], based on the Morgan fingerprint [34]. Second, we developed three graph neural network-based molecular feature extraction models by integrating the optimal machine learning classifier GBDT, and binary classification models were constructed. Third, five-fold cross-validation was used to select the most suitable GNN model, which is DMPNN in this study. Finally, DMPNN + GBDT ensemble feature extraction and a target prediction model were used to predict drug multitargets of HIV-1 and HBV-related targets. The detailed results are shown below:

### 2.1. Experimental Data Sets

In this study, a total of seven targets related to HIV-1 and five targets related to HBV were obtained from the ChEMBL database [35] (version 32, https://www.ebi.ac.uk/chembl/ (accessed on 1 May 2022)) (shown in Table 1). The number of active compounds for each target is shown in Figure 2 (the criteria of “active” and “inactive” associations are shown in Section 4). The number of active compounds acted on HIV-1 and HBV targets were 13,627 and 2093, respectively, the number of inactive compounds related to HIV-1 and HBV targets were 4142 and 1120, respectively. Each of these 12 data sets was randomly divided into the training, validation, and test data sets by proportions of 0.8, 0.1, and 0.1, respectively. Table 1 shows the number of compounds in the training and test sets from the 12 data sets.

### 2.2. Comparison of the Classification Performance of Different Machine Learning Methods

The most popular strategy for molecular property prediction is to calculate the Morgan fingerprints of the given molecules and then use machine learning classifiers to train the prediction model. This study first compared the classifying performance of these traditional supervised learner methods: SVM, RF, NB, XGBoost, and GBDT, proving that GBDT outperformed them on all of the 12 data sets (100%) in the area under receiver operating characteristics curve (AUC) values, and on 11 out of 12 (91.67%) for the F1 metric, which is defined as the harmonic mean of precision (PPV) and recall (TPR) for the active class (Figure 3), which would be used to replace the output layers of GNNs for prediction.

### 2.3. Performance Evaluation of Feature Extraction between Graph Representations and Supervised Learners

In our study, three kinds of GNNs were used to construct graph representations to extract molecular features of compounds; subsequently, the output layers were further improved by a selected supervised learner GBDT for classification. Consequently, three different binary classification ensemble models (GCN + GBDT, GGNN + GBDT, and DMPNN + GBDT) were built for each of the target-associated compound data sets. All three GNNs achieved the satisfied convergence results, converging within 200 epochs on both training and validation samples, shown in Figure 4. Moreover, the 5-fold cross-validation tasks in the context of 12 single-target binary classifications were used to compare the performance of these 3 ensemble models. For each algorithm and target, hyperparameters were optimized from an exhaustive search, and detailed information is shown in the Section 4.

The performance of different kinds of classifiers in the 12 single-target predictions is detailed in Figure 5. Overall, the ensemble models performed better in the single-target prediction task than the simplex graph neural models or machine learning methods. Furthermore, the DMPNN-based ensemble model outperformed the other two GNNs on 10 out of the 12 data sets (83.3%) in the AUC values, and also 10 out of the 12 (83.3%) in the F1 metric, suggesting that it has more significant extracting ability for the current sets of HIV-1 and HBV target studies. Thus, ensemble graph-based models using integrated features can effectively improve the extraction efficiency of molecular features. Meanwhile, it is essential to evaluate the performance of different graph neural network-based feature extraction algorithms on a specific prediction problem before further analysis.

### 2.4. Multitarget Prediction

In this study, we aimed to exploit drugs with dual antiviral agents to HIV/HBV coinfection. Correspondingly, based on the previous cross-validated predictions of 12 individual binary classifiers, the most efficient predicting ensemble model, DMPNN + GBDT, was adopted to finish HIV-1 and HBV multitarget prediction. However, each classifier was trained and tested on compound data sets that do not coincide completely. Because some compounds’ response information to certain targets does not exist in the ChEMBL database, we had to mark them as no response. Therefore, such negative labels do not indicate truly negative cases. For this reason, we used 5 compound sample data sets, in which elements containing no less than 3 active interactions between compounds and targets for all the 12 targets. Their sample sizes were between 110 and 150 with no more than 20 repeated compounds among them. For each sample, the selected ensemble graph neural network-based prediction model from the preceding step was used to predict multitargets on the five compound samples. In addition, this model was re-trained on the whole compound data sets excluding these five data samples. The distribution of the performance of DMPNN + GBDT across these five compound samples is shown in Figure 6. By reason of in a practical medicinal chemistry application, the correct identification of active compounds is often more important than the identification of inactive ones. In this case, metrics considering the correct identifications of the active compound–target associations, negative predictive value, and the false negative rate (*FNR)* were calculated: *TPR*, *NPV*, and *FNR* ($TPR=\frac{TP}{TP+FN}$, $NPV=\frac{TN}{TN+FN}$, $FNR=\frac{FN}{TP+FN}$). The ensemble model can achieve relatively high *TPR* and *NPV* values, meaning there is a high accuracy of detection for positive interactions. Additionally, they show extremely low *FNR* values, meaning nearly no omission of true active interactions in our prediction.
Case 1: Retrospective polypharmacology prediction of known HIV-1/HBV drugs

To explore the potential multiple targets of the known HIV-1/HBV drugs, a DMPNN + GBDT prediction model based on the 12 key targets related to HIV-1 and HBV was used to predict potential multiple bioactivities among them for the approved 22 HIV-1 drugs (abacavir, emtricitabine, lamivudine, viread (TDF), zidovudine, doravirine, efavirenz, etravirine, nevirapine, rilpivirine, atazanavir, darunavir, fosamprenavir, ritonavir, tipranavir, enfuvirtide, maraviroc, catotegravir, dolutegravir, raltegravir, fostemasavir, cobicistat) and 8 HBV drugs (baraclude, tenofovir, viread, lamivudine, hepsera, adefovir, tenofovir alafenomide (TAF), peginterferon alfa-2b). Figure 7 shows the multitarget distribution of these drugs, identified by our ensemble model. To verify the prediction results, the predicted targets were analysed using the PubChem BioAssay database [36], eight approved drugs and the corresponding experimental records are shown in Table 2.
Case 2: Multitarget prediction of new compounds

One of the most important applications of establishing a new prediction model is drug discovery. In our research, we aimed to find twin target directional drugs for HIV-1/HBV coinfection. Therefore, after the above verification of a potential new pharmacy of known approved clinical medicine, a set of ten compounds with known activity targeting at least one of the seven targets (PARB, CD299, CXCR4, IN, PR, RT) that was not involved in the training of the DMPNN + GBDT model was retrieved from the PubChem database. Among these prediction results, six compounds (the 2D structures and bioactivity data from ChEMBL data set are shown in Figure 8) were predicted to be simultaneously active against multiple HIV-1 and HBV targets, as shown in Figure 9. Moreover, some new predictions have been verified from previous research, such as CHEMBL38700 (name: Elvucitabine), which has been tested for the ability to inhibit HBV DNA synthesis using human hepatablastoma cell line HepG2 2.2.15 [52]; we also found the evidence of the inhibitory activity against growth of HIV-infected DLD-1 cell line and Hela CD4 cells with activity < 1 µm [53,54]. Moreover, in order to verify more detailed binding modes of these six compounds’ targets on the predicted targets, the molecular docking analysis provided more detailed inspection and evidence of the multibinding relationships of compounds and HIV-1/HBV targets (Figure 10), where all of these compounds occupied one stabilized hydrophobic pocket or hydrogen bonds formed by the targeting proteins, with detailed results shown in Figure 11.

CHEMBL38700 was predicted to be an HBV CD299 and HIV-1 CXCR4 PR in inhibitor, as shown in Figure 11a–c. The carboxyl group of CHEMBL38700 forms hydrogen bonds with the main chain carbonyl oxygen of residues GLN-276, GLY-277, TRP-339 and GLV-352 in target CD299 (Figure 11a) and ASP-30 and ASP-29 in target PR (Figure 11c). CHEMBLE38700 was occupied a hydrophobic pocket formed by residues ARG-1014, TYR-1018, and TYR-1024 in target CXCR4 (Figure 11b).

CHEMBL1223975 and CHEMBL1223977 were predicted to target with IN and PARB. CHEMBL1223975 occupied a hydrophobic pocket formed by residues LYS-127, LEU-101, and ILE-135 in target IN (Figure 11d), and linked by hydrogen bonds with residues ASN-142, THR-62, and ASP-141 in target PARB (Figure 11e). CHEMBL1223977 was occupied a hydrophobic pocket formed by residues F121, W131, K136, and L101 in target IN (Figure 11f), and forms hydrogen bonds with the residues ARG-64, and ASP-141 in target PARB (Figure 11g).

CHEMBL1223979 was predicted to target CXCR4 and PARB. It occupied the hydrophobic pocket formed by the residues ARG-30, TRP-94, HIS-281, ASP-262, VAL-196, and GLN-200, and also linked by a hydrogen bond formed by residue ASP-262 in target CXCR4 (Figure 11h). The interaction with target PARB was formed by the hydrogen bonds directly with residues GLU-146, ASN-143, ASP-143, and ALA-143 (Figure 11i). Similarly, CHEMBL2092833 was predicted to be active on targets CD299, CXCR4, and IN. While CHEMBL2092835 was predicted to be active on targets CD299, CXCR4, and RT. The detailed docking information is shown in Figure 11j–n.

## 3. Discussion

Approximately 2 billion people worldwide have been infected with hepatitis B virus (HBV), of whom more than 350 million suffer from chronic infection and between 500,000 and 700,000 people die from HBV infection each year. Additionally, as of 2019, an estimated 38 million people worldwide were living with HIV, including 1.8 million children under age 15. In 2019, nearly 690,000 people worldwide died from AIDS-related illness. Because of the similar transmission routes of HIV and HBV, the two blood-borne viruses, the high incidence of coinfection leads to major global health problems. In contrast to single infections of these viruses, coinfections can lead to a variety of liver-related diseases, nonliver organ dysfunction, and death. Treatment of coinfected patients is complicated due to the side effects of antiviral drugs, resulting in drug resistance, liver toxicity, and lack of an effective response. Moreover, coinfected individuals must be treated with multiple drugs at the same time, which results in the complex diagnosis, treatment, and control of HIV and HBV coinfections. Therefore, the research and development of novel drugs with multiple targets is an urgent problem.

However, traditional drug discovery methodology using experiments to screen candidate drugs is expensive and time consuming. With the development of machine learning and deep learning algorithms, rapid innovations in virtual screening have been realized. One of the most important procedures for the application of machine learning models for large-scale compound screening is molecular feature extraction. Graph neural networks (GNNs) opened up breakthroughs for learning interatom connections because of their representation ability for the spatial graph structures of molecules.

In this study, we compared three kinds of graph neural networks for their ability to extract molecular features by replacing the output layers of these neural networks with one optimal supervised learning algorithm, GBDT. The ensemble model DMPNN + GBDT was selected for HIV-1/HBV multitarget fishing based on the combination of 12 binary classifiers. The proof-of-principle study demonstrated that the integration of DMPNN and GBDT may efficiently improve the prediction performance regarding HIV-1 and HBV targets compared with single machine learning or single neural network algorithms. Furthermore, 22 approved HIV-1 drugs and 8 approved HBV drugs were predicted to have potential multitargets, with 8 of them validated by references.

Furthermore, to verify our target predictions of the ten new compounds, molecular docking simulations were adopted to provide a detailed inspection of the binding mode for each compound, where six compounds were predicted to have twin relationships with HIV and HBV targets, which have been verified by the molecular docking algorithm. Thus, the results showed that it is possible to design specific inhibitors that target HIV-1 and HBV simultaneously. In addition, our ensemble model has the possibility to effectively detect multitargeted co-inhibitors of HIV/HBV coinfection and has potential applications for virtual screening multitarget drugs for the treatment of other complex diseases.

## 4. Materials and Methods

### 4.1. Data Set Preparation

The ChEMBL Database [35] (version 32) and Therapeutic Target Database (TTD) were used to select the clinical targets related to HIV-1 and HBV. The keywords “HIV-1”, “Human Immunodeficiency Virus type-1”, “HBV”, and “Hepatitis B Virus” were used as retrieval conditions. Then, the search results were further filtered if the number of compounds related to the target was less than 30. Meanwhile, compounds with a quantitative measure of biological activity (IC_50_, EC_50_, K_i_, or K_d_) lower than or equal to 10 μM were labelled “active”; in contrast, those with a quantitative measure of biological activity higher than 10 μM were labelled “inactive”.

### 4.2. Molecular Feature Extraction by Graph Neural Network

The Simplified Molecular Input Line Entry System56 (SMILES) [55] of compounds was downloaded from the ChEMBL database, and the tool RDKit [56] was used to process these SMILES encoding compounds to obtain molecular graphs and Morgan finger prints [34]. Then, three graph neural network models (GCN, GGNN, DMPNN) were adopted to learn the representation of the molecular structures, where each graph is composed of nodes and edges. Nodes are described by the type of atom, atom elements, number of additional atoms, number of valence electrons, aromatic properties, and other properties. The adjacency matrix represents the connectivity between pairs of atoms, regardless of single or double bonds. In the graph convolutional neural network (GCN), the states of the graph nodes are updated using the embedding method: $h_{i}^{t}=U(h_{i}^{t-1},m_{i}^{t})$, where the ith node was updated by the previous node state $h_{i}^{t-1}$ with the message state $m_{i}^{t}$. The gated graph neural network (GGNN) utilizes the gate recurrent units (GRUs) [57] in the propagation step. Furthermore, the directed MPNN contains three major operations: message passing, node update, and readout:$$\begin{array}{l} m_{i}^{t}=\sum_{j\in N(i)}M_{t}(h_{i}^{t},h_{j}^{t},e_{ij})\\ h_{i}^{t+1}=U_{t}(h_{i}^{t},m_{i}^{t}) \end{array}$$
where $M_{t}$ is the message function, $U_{t}$ is the node update function, and N(i) is the set of neighbors of node i in graph G. The readout phase uses a readout function R to calculate the feature vector of the whole graph:$$\hat{y}=R(h_{v}^{T}|v\in G)$$

The message function, update function, and the readout function are all differentiable. R operates on the set of states of nodes and is insensitive to the arrangement of nodes, which lead to MPNN invariant to graph isomorphism. Rather than using messages associated with vertices (atoms) in the generic MPNN, Directed MPNN (DMPNN) uses messages associated with directed edges (bonds) in the molecules, which would avoid the messages being totters, that is, to prevent unnecessary loops in the message passing trajectory [58]. In our study, DMPNN demonstrated superior performance in molecular property extraction.

### 4.3. Multitarget Prediction Model

In this study, to generate a multitarget classifier, three graph neural network-based ensemble models integrating graph representation and Morgan representation of molecular structures were evaluated in 12 binary classifier data sets. The original output layer of each GNN was replaced by the gradient boosting decision tree (GBDT), which achieved the best prediction performance among all the other supervised learners: SVM, RF, and XGBoost. Then, by combining these 12 binary classifiers, the multitarget predictor for identifying HIV-1/HBV target-compound associations was generated, which is named DMPNN + GBDT.

### 4.4. Machine Learning Methods

To select the cooperation of the graph neural network in the collaborating duets, six kinds of machine learning algorithms were evaluated for the performance of the binary-target classification task: random forest (RF), support vector machines (SVM), naive Bayes (NB), gradient boosting decision tree (GBDT), and extreme gradient boosting algorithm (XGBoost). The hyperparameter tuning of all machine learning methods was optimized using 5-fold cross-validation in an exhaustive search over a limited hyperparameter space. In RF classification, for a given sample, each decision tree predicts a label, and the final prediction of the sample is the label of most tree predictions, in which the number of decision trees was fixed to 1000. SVM classification constructed the best separation of two classes by maximizing the distance (binary) between instances that belong to different classes. For this algorithm, a “cost” hyperparameter used to control the errors was optimized by the candidate values 0.01, 0.1, 1, 10, and 100. GBDT is a member of the ensemble learning boosting family, which iterates using a forward distribution algorithm. Assuming the strong learner from the previous iteration is $f_{t-1}(x)$, the loss function is $L(y,f_{t-1}(x))$, and the goal of this iteration is to find a decision tree learner $h_{t}(x)$, minimizing the loss function $L(y,f_{t}(x))=L(y,f_{t-1}(x)+h_{t}(x))$. For this algorithm, the number of decision trees was fixed to 1000; the maximum depth of the decision tree was selected from 3, 5, 10; the learning rate was sampled from (0, 1); and the minimum number of samples to split an internal node was a candidate from 2, 3, 4, and 5. Finally, 3 XGBoost parameters were tuned, i.e., the learning rate, max depth, and minimum child weight from 5 candidate values 0.01, 0.05, 0.1, 0.15, and 0.2.

### 4.5. Performance Metrics

This study evaluated the proposed framework DMPNN + GBDT with prediction performance with the other two graph neural networks in the single-target prediction validation. The F1 score and AUC, were adopted to assess the performance of each model. The F1 score is the harmonic mean of precision (*PPV*) and recall (*TPR*), the AUC score is defined as the area under the receiver operating characteristics (ROC) curve for the classification performance metric. The ROC curve is defined as the connected curve of the points true positive rate (*TPR*), and false positive rate (*FPR*).
$$F1=2\times \frac{TP}{2TP+FP+FN}$$
$$\begin{array}{l} TPR=\frac{TP}{TP+FN}\\ FPR=\frac{FP}{FP+TN} \end{array}$$
where *TP* means true positives, *TN* means true negatives, *FP* means false positives, and *FN* means false negatives, with positive and negative referring to active and inactive compound labels, respectively.

To assess the global performance of our multitarget classifier combined with 12 single-target binary classifiers in the HIV-1/HBV target prediction task, the capability of identifying the known active compounds among the known compound target associations *NPV*, *TPR*, and the ability of no missing known interactions *FNR* were used as metrics.
$$\begin{array}{l} NPV=\frac{TN}{TN+FN}\\ TPR=\frac{TP}{TP+FN}\\ FNR=\frac{FN}{TP+FN} \end{array}$$

### 4.6. Data Imbalance

As shown in Figure 2, the data sets used in this study are imbalanced. We adopted the cost sensitive learning method to increase the importance of certain categories in the classification training tasks by punishing classification errors in these categories specially, by adjusting the parameters in machine learning methods. For XGBoost, the parameter “scale_pos_weight” was set to the ratio of numbers of negative cases and positive cases. As for SVM and RF, parameter “class_weight” was set to “balanced”. There is no technical parameter for balancing data in the NB algorithm. However, for decision tree models, which use partition rules based on class variables to create classification trees, they can achieve good performance by separating samples from different categories forcibly. Therefore, in GDBT, we set the parameter “sample_weight” according to the proportion of negative samples and positive samples in different data sets, resulting in reducing the impact of large sample sets artificially.

## 5. Conclusions

In this article, we proposed one graph neural network-based predicting model by integrating one efficient supervised learning algorithm that is an excellent implementation of the gradient boosting strategy, GBDT. By combing 12 binary optimal classification data sets, 1 multiple target prediction model was constructed. In order to evaluate the performance of our multitarget prediction ensemble model, five external data sets were constructed for the prediction evaluations, all of which achieved the satisfied PPV and TPR, meaning the relatively high accurate prediction of potential targets on HIV-1 and HBV. Furthermore, our research gave the retrospect of known HIV-1 and HBV drugs for detecting potentially possible targets, also explored new compounds for their twin targets of HIV-1 and HBV, resulting in six new potential compounds for treating co-infection of HIV-1 and HBV. In conclusion, the neural network model used in this study could effectively identify co-inhibitors for HIV-1 and HBV, and have the potential application for virtual screening of multitarget drugs for the treatment of HIV/HBV coinfection.

## Figures and Tables

**Figure 1 ijms-24-07139-f001:**
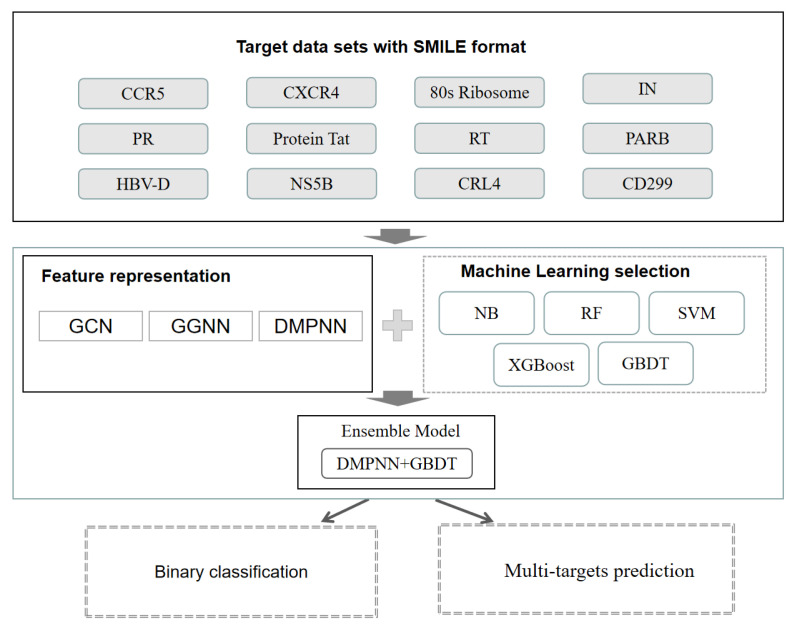
Workflow of our study.

**Figure 2 ijms-24-07139-f002:**
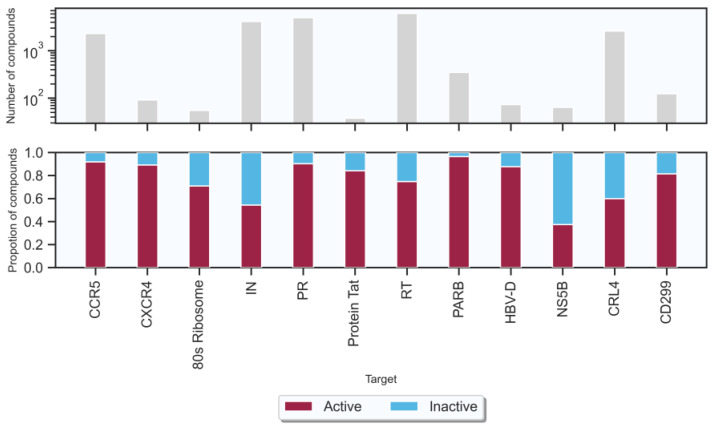
Size and composition of the target-associated compound data sets. The number of compounds in the upper panel is displayed on a logarithmic scale.

**Figure 3 ijms-24-07139-f003:**
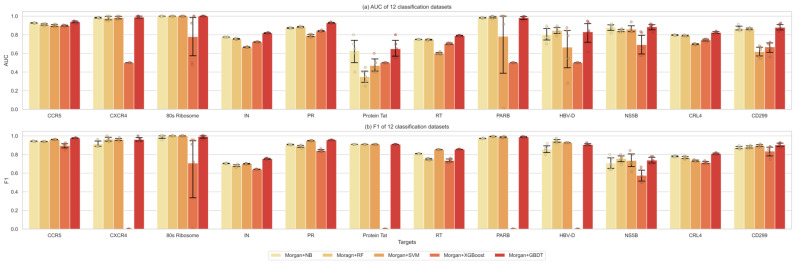
The performance comparison of different supervised learners. (**a**): the AUC values. (**b**): F1 metrics.

**Figure 4 ijms-24-07139-f004:**
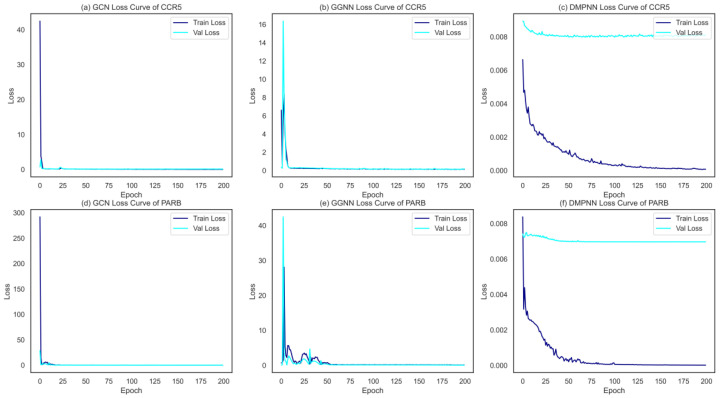
The loss curves of the three GNNs on one HIV-1 target (**a**–**c**), and one HBV target (**d**–**f**).

**Figure 5 ijms-24-07139-f005:**
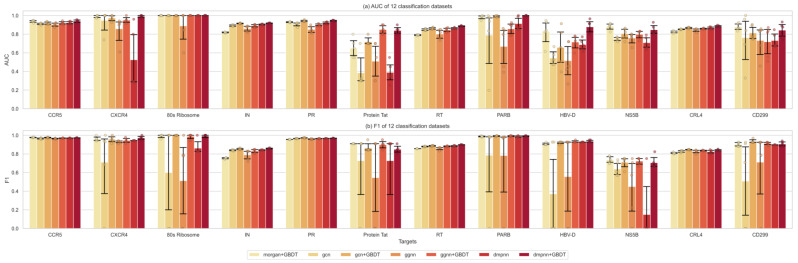
The performance comparison of different prediction models: morgan + GBDT, GCN, GCN + GBDT, GGNN, GGNN + GBDT, DMPNN, and DMPNN + GBDT. (**a**): the AUC values. (**b**): F1 metrics.

**Figure 6 ijms-24-07139-f006:**
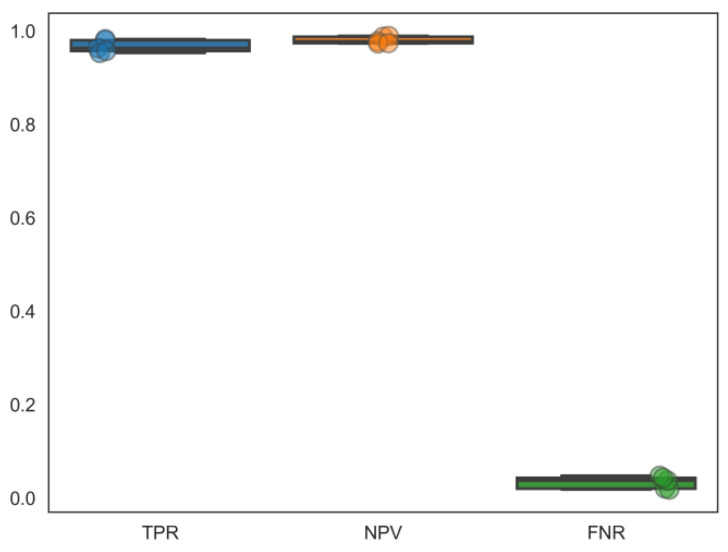
Performance metrics (*TPR*, *NPV*, *FNR*) to evaluate our model’s applicability in multitargets prediction. Different colors means different performance evaluation.

**Figure 7 ijms-24-07139-f007:**
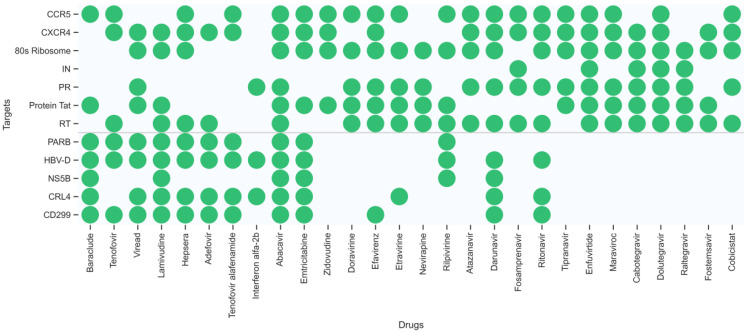
The multitarget distribution of 30 approved drugs identified by our ensemble model.

**Figure 8 ijms-24-07139-f008:**
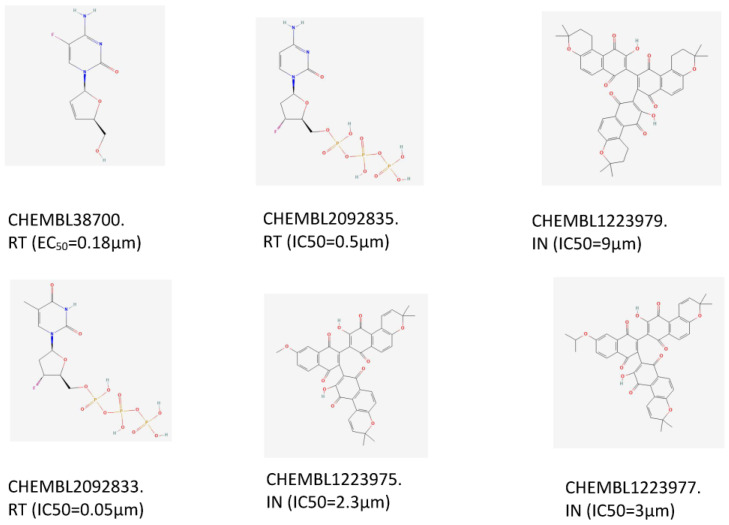
The 2D structures and bioactivity data of 6 compounds.

**Figure 9 ijms-24-07139-f009:**
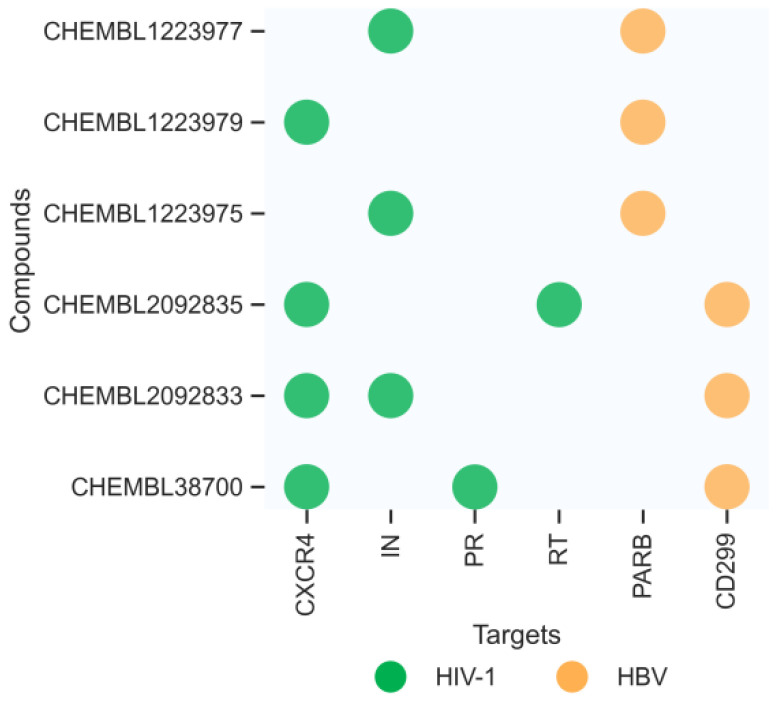
The distribution of multitargets predicted by our ensemble model for the six new compounds.

**Figure 10 ijms-24-07139-f010:**
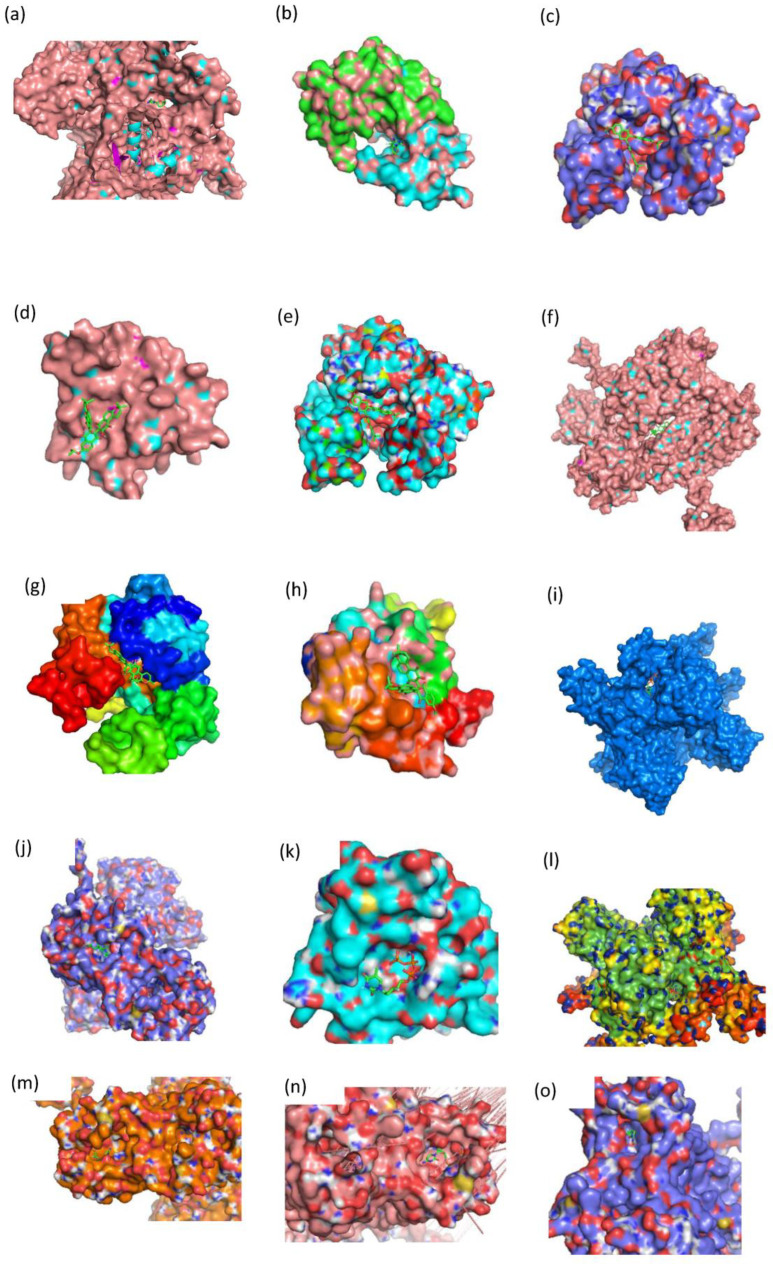
The binding modes of six compounds in their respective binding sites. (**a**): CHEMBL38700-CD299. (**b**): CHEMBL38700-PR. (**c**): CHEMBL1223975-PARB. (**d**): CHEMBL1223977-IN. (**e**): CHEMBL1223977-PARB. (**f**): CHEMBL1223979-CXCR4. (**g**): CHEMBL1223979-PARB. (**h**): CHEMBL1223985-IN. (**i**): CHEMBL2092833-CD299. (**j**): CHEMBL2092833-CXCR4. (**k**): CHEMBL2092833-IN. (**l**): CHEMBL2092835-CD299. (**m**): CHEMBL2092835-CXCR4. (**n**): CHEMBL2092835-RT. (**o**): CHEMBLE38700-CXCR4.

**Figure 11 ijms-24-07139-f011:**
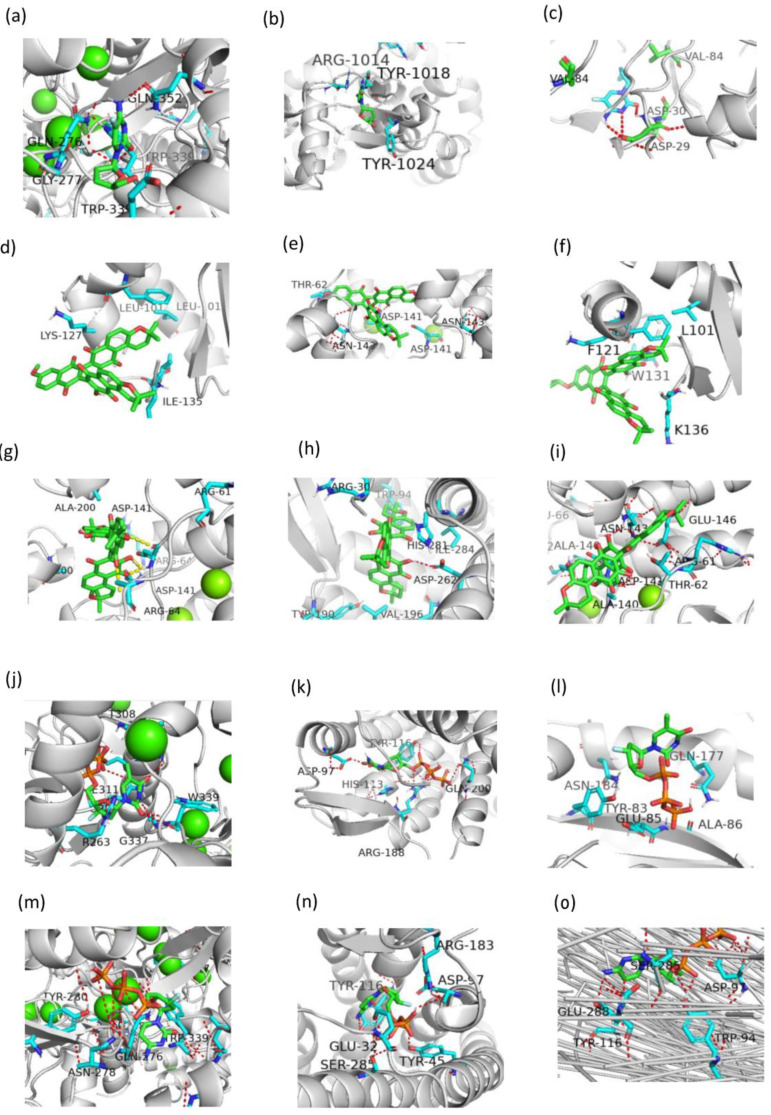
The detailed binding modes corresponding to Figure 10. (**a**): CHEMBL38700-CD299. (**b**): CHEMBL38700-PR. (**c**): CHEMBL1223975-PARB. (**d**): CHEMBL1223977-IN. (**e**): CHEMBL1223977-PARB. (**f**): CHEMBL1223979-CXCR4. (**g**): CHEMBL1223979-PARB. (**h**): CHEMBL1223985-IN. (**i**): CHEMBL2092833-CD299. (**j**): CHEMBL2092833-CXCR4. (**k**): CHEMBL2092833-IN. (**l**): CHEMBL2092835-CD299. (**m**): CHEMBL2092835-CXCR4. (**n**): CHEMBL2092835-RT. (**o**): CHEMBLE38700-CXCR4.

**Table 1 ijms-24-07139-t001:** Detailed statistical analysis of the 15 data sets.

Object	Target	Total	Training Set			Test Set		
			Active	Inactive	Total	Active	Inactive	Total
HIV-1	CCR5: C-C chemokine receptor type 5	2296	1478	129	1607	633	56	689
	CXCR4: C-X-C chemokine receptor type 4	92	57	7	64	25	3	28
	80s Ribosome	55	27	11	38	12	5	17
	IN: Integrase	4161	1584	1328	2912	679	570	1249
	PR: Protease	5005	3162	341	3503	1356	146	1502
	Protein Tat: Human immunodeficiency virus	38	22	4	26	10	2	12
	RT: Reverse transcriptase	6122	3207	1078	4285	1375	462	1837
HBV	PARB	350	237	8	245	101	4	105
	HBV-D: HBV genotype D	73	45	6	51	19	3	22
	NS5B: RNA-dependent RNA polymerase	64	16	28	44	8	12	20
	CRL4: E3 ubiquitin ligase	2614	1096	733	1829	470	315	785
	CD299: Core antigen of hepatitis	124	70	16	86	31	7	38

**Table 2 ijms-24-07139-t002:** Known experimental verification of approved HIV-1 and HBV drugs.

Drugs	Experimental Factor Ontology (EFO) Terms	Max Phase for Indication *	References
tenofovir	HIV-1 infection	3	[37]
tenofovir	Hepatitis B virus infection	3	[38], FDA
hepsera	Chronic hepatitis B virus infection	4	[39],
hepsera	Hepatitis B virus infection	4	[40], FDA
hepsera	HIV infection	3	[41]
adefovir	HIV infection	1	[42]
adefovir	Hepatitis B virus infection	3	[40], FDA
tenofovir alafenamide	HIV-1 infection	4	[43]
tenofovir alafenamide	Hepatitis B virus infection	4	[44], FDA
interferon alfa-2b	HIV-1 infection	3	[45]
interferon alfa-2b	Hepatitis B virus infection	4	[46], FDA
abacavir	HIV-1 infection	4	[47], FDA
emtricitabine	HIV-1 infection	4	[48], FDA
emtricitabine	Hepatitis B virus infection	3	[49]
ritonavir	Hepatitis B virus infection	2	[50]
ritonavir	HIV-1 infection	4	[51], FDA

*: Denotes the maximum phase of development for the compounds across all indications.

## Data Availability

All Python codes of our models in this manuscript are available: https://github.com/Yswangustb/Multitarget-prediction, accessed on 1 April 2023.

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
