# Peer review of "HIV-1/HBV Coinfection Accurate Multitarget Prediction Using a Graph Neural Network-Based Ensemble Predicting Model"

_ijms, 2023, doi:10.3390/ijms24087139_

Round 1
Reviewer 1 Report
Dear colleagues! You research is very interesting and describes a new promising approach to find new drugs through quite a reliable procedure.
The only comment is about the molecular docking results and presentation. The figures are impressive, but the detailed interaction between ligands and receptors can not be evaluated from them. It is just my personal experience that speaks here, but from practical point of view it is more important to have a look at the possible non-covalent ligand-receptor bonds. Would you please provide the picture with zoomed binding centers and interactions between ligangs with essential residues?
Reviewer 2 Report
Wang et. al. in the present work show the use of machine learning in drug discovery. They used predictive model to identify molecules which are "active" against multiple targets from HIV and HBV, thus overcoming several issues with current method of disease treatment. Overall it is an interesting effort to demonstrate the application of continuously evolving field of machine learning and neural networks. I suggest following changes:
1. The article can benefit from extensive rewriting and reorganization of presentation of details. The introduction is unnecessarily long on discussing HIV/HBV and remarkably absent on introducing at least two algorithms which were finally used in the paper, namely DMPNN and GBDT.
2. By reading workflow, I got idea that DMPNN follows GBDT but it is the other way in the implementation. The figure 1 is misleading. The authors can explain how the data flows through the architecture. Also, the authors can discuss what is the output of DMPNN (can it be related to molecular properties or is it just internal representation unreadable for humans) which is fed to GBDT. A figure summarizing data flow will be helpful.
3. Figure 2b is redundant. A column of total molecules added to Table 1 will suffice.
4. The data set is heavily imbalanced. How did the authors overcome it?
5. The section on multi target prediction has to rewritten. It is not clear how the data is collected for overcoming false negatives due to lack of data on molecules with common targets from both HIV and HBV. What do the authors mean by randomly generated in following?
".....with all 12 targets were randomly generated...."
6. Does this method suggested by authors help in generating new drugs? Or they help in repurposing old molecules for new targets?
Minor comment:
Please make sure acronyms are defined at their first usage. MPNN is never defined. And HAART is for Highly Active AntiRetroviral Therapy instead of Highly effective combined antiretroviral therapy.
